# In-Depth Analysis of an Obligate Anaerobe *Paraclostridium bifermentans* Isolated from Uterus of *Bubalus bubalis*

**DOI:** 10.3390/ani12141765

**Published:** 2022-07-09

**Authors:** Purva Gohil, Kajal Patel, Srushti Patel, Ramesh Pandit, Vishal Suthar, Srinivas Duggirala, Madhvi Joshi, Deepak Patil, Chaitanya Joshi

**Affiliations:** 1Gujarat Biotechnology Research Centre, Gandhinagar 382011, India; purvagohil5@gmail.com (P.G.); kajalpatel302@gmail.com (K.P.); panditramesh.gbrc@gmail.com (R.P.); madhvimicrobio@gmail.com (M.J.); director.gbrc@gmail.com (C.J.); 2Department of Microbiology, Gujarat Vidyapith, Gandhinagar 382320, India; srushtipatel.micro@gmail.com (S.P.); dsrinivasmurty@gmail.com (S.D.); 3Directorate of Research, Kamdhenu University, Gandhinagar 382010, India; db1608@gmail.com

**Keywords:** *Bubalus bubalis*, endometritis, *Paraclostridium bifermentans*, genomic research, toxin, virulence

## Abstract

**Simple Summary:**

Non-specific uterine infections in bovine (uterine line inflammation) are a significant issue for the dairy industry. These infections are responsible for significant financial losses all over the world. *Paraclostridium bifermentans* is an obligate anaerobe, gram-positive rod-shaped bacteria belonging to the *Clostridia* class and the *Peptostreptococcaceae* family produces endospores. This bacterium has the ability to infiltrate bovine uterine endometrial epithelial cells and cause infection in the endometrium epithelial cells. Our study found that an examination of a buffalo uterus with yellowish purulent discharge reported the presence of pathogenic bacteria *Paraclostridium bifermentans*, where its genomic characterization, substrate utilization, and antibiotic susceptibility potentiality was studied. This discovery indicates the presence of virulence genes as well as pathogenic features. This is the first report of *P. bifermentans* from the bovine uterus environment.

**Abstract:**

Chronic non-specific contamination of the reproductive tract in animals is a major issue during early postpartum, natural coitus, or artificial insemination. Uterine infection is one of the major concerns reducing fertility, production loss, and early culling of the animals. Therefore, the aim of this study was to identify any novel bacterium if present in the uterine environment of *Bubalus bubalis* causing infections. A strictly anaerobic bacterial strain designated as *Paraclostridium bifermentans* GBRC was isolated and characterized. Bacterium was found to be Gram positive moderate rod with motility. The optimum growth was observed at 40 ± 2 °C. The pathogenic characteristics of the GBRC strain, such as hemolysis, gelatin hydrolysis, and the production of volatile sulfur compounds, were similar to those seen in the epithelial layer invading pathogenic strains. Assembled genome size was 3.6 MB, with 78 contigs, and a G + C content of 28.10%. Furthermore, the whole genome sequence analysis confirmed the presence of genes encoding virulence factors and provided genomic insights on adaptation of the strain in the uterine environment. Based on the phenotypic and genetic differences with phylogenetic relatives, strain GBRC is proposed to represent a first reported species of the genus *Paraclostridium* with potential pathogenic character, from the buffalo uterine environment. This study analysis of the GBRC strain serves as a key reference point for the investigation of potential pathogenic strains that may cause endometritis and metritis in bovine.

## 1. Introduction

The *Bubalus bubalis* species is a significant participant in the Indian dairy sector. Optimum fertility of dairy buffaloes is inevitably vital for economically sustainable dairy production. Therefore, maintaining the reproduction performance at an optimal level is a priority. The non-specific uterine dysfunction causes a reduction in reproductive efficiency, animal production, as well as increasing the diagnostic and treatment costs, which ultimately results in the culling of the animal from the herd. Farmers continue to bear the brunt of this issue, which is one of their most expensive problems. Many opportunistic organisms from natural flora or from the environment may infiltrate the reproductive system during postpartum, particularly in buffaloes with low immunity, causing infertility in buffaloes. Apart from non-specific contamination during calving, other pathogens can gain access to the reproductive organs of the buffaloes during natural service or artificial insemination via contaminated sources [1,2].

Many studies and reviews have reported metritis, clinical endometritis, and subclinical endometritis ranging from 8 to 40%, 12 to 35% and 7 to 35%, respectively in dairy bovines [2,3,4,5]. In Indian cows and buffaloes, the reported ranges for metritis, clinical endometritis, and subclinical endometritis are from 7 to 40%, 8 to 38% and 13 to 27%, respectively [6,7,8]. The pathogens such as *Trueperella pyogenes*, *Escherichia coli*, *Prevotella melaninogenica*, and *Fusobacterium necrophorum* are recognized as uterine pathogens responsible for metritis, endometritis, and subclinical endometritis. Many studies and reviews have reported and described common opportunistic pathogens and their plausible mechanisms of action in depth [6,8,9]. Nonetheless, the literature on the connections of these four pathogens with uterine disease has not always been consistent, especially with regards to the involvement of *Escherichia coli* and *Prevotella melaninogenica* [9].

Much has been undertaken to explore uterine normal flora, opportunistic organisms, and infections; nevertheless, there are still species that have not been identified that need to be investigated. There are still many unexplored/unknown pathogens that need to be investigated for their involvement in the development of uterine endometrial disease. Therefore, the aim of this study was to identify any novel bacterium if present in the uterine environment of *Bubalus bubalis* causing infections.

## 2. Materials and Methods

### 2.1. Collection of Samples

Our team routinely inspects the uterine organ of adult buffalos at the slaughterhouse for education purposes. In one case, a buffalo (apparently looking multiparous) uterus with unidentified reproductive history and yellowish purulent discharge from the uterus was taken into considerations. Both horns were symmetric, flaccid, and revealed fluid on palpation. The cervix had a cylindrical shape and fully revealed an involuted uterus with mild inflammation. The right ovary had a corpus luteum and multiple small and medium follicles on the left ovary. NS (sterile normal saline) (0.85 percent NaCl) was used to clean the uterus’ surface aseptically shortly after slaughter at the Ahmedabad Municipal Corporation slaughterhouse in Gujarat. Within 3 h, the slaughtered uterus was transported to the laboratory at 4 °C using a transport sipper. The samples were obtained aseptically from the uterus’s left and right horns by flushing using a sterile 18 G needle and a 20 mL syringe in a sterile environment. For flushing, uterine horns were punctured from the uppermost area near the body of the uterus. In both horns, 40 mL NS was introduced and recovered back in the syringe within 30–40 s. For both horns, separate sterile needles and syringes were used. The flushed NS was collected in the 10 mL vacutainers. When collecting the samples, every precaution, including aseptic techniques, was followed.

### 2.2. Bacterial Isolation and Culture Condition

The anaerobic culture techniques as described by Hungate was used for the enrichment of the anaerobic bacteria [10]. The flushed NS sample was enriched in modified Mah et al. medium [11] and Anaerobic Enrichment Medium supplemented with peptone, and yeast extract. To isolate obligate anaerobes, serial dilutions [diluent contained: 0.5% NaCl and 0.2% Cysteine HCl, pre-gassed with N2:H2 (80:20)] were prepared from the enriched samples and 0.1 mL from each dilution was streaked on solid medium in anaerobic glove box (Don Whitely, Bingley, UK) and incubated in anaerobic glove box maintaining strict anaerobic conditions by purging N_2_: CO_2_ (80:20). Plates were incubated at 40 ± 2 °C, and colony morphology was noted. Subsequently, three sequential transfers were performed for each well-isolated colony to ensure the purity of culture.

### 2.3. Morphological and Biochemical Characterization

The isolate’s growth, colony characteristics, and morphology were studied using the methods outlined in Bergey’s Manual [12]. General characteristics of *P. bifermentans* JCM 1386T [13]; *P. benzoelyticum* JC272T [14]; *P. bifermentans* subsp. *muricolitidis* [13]; and *P. bifermentans* GBRC were compared. To evaluate the isolate’s substrate utilization ability, the carbon and nitrogen source utilization ability of the isolate was studied using the AN-Biolog^®^ microplate technique. After 72 h of incubation, the OD595 was determined using a microtiter plate reader. Antibiotic sensitivity test of the isolate against 16 antibiotics was carried out using HiMedia 8 *G-I-plus.

### 2.4. DNA Extraction and Library Preparation

For DNA extraction, isolate grown in roll tube was washed with Phosphate Buffered Saline (PBS) and CO₂ gas purged at low pressure at the same time. PBS containing suspended cells was collected using a sterile syringe and transferred into falcon tubes. Genomic DNA was extracted as described by Wilson [15]. The quality (260/280 and 260/230 ratio) of the DNA was evaluated using a QIAexpert system (QIAGEN, Hilden, Germany) as well as with electrophoresis on 1% agarose gel. DNA quantification was carried out using Qubit dsDNA HS Assay Kit with Qubit 4.0 (Thermo Fisher Scientific, Waltham, MA, USA). Before going for the whole genome sequencing, the culture was identified using 16S rDNA gene sequencing. For library preparation, Ion Plus Fragment Library Kit (Thermo Fisher Scientific, Waltham, MA, USA) was used. The quality of the library was assessed using Bioanalyzer (Agilent, CA, USA) and the library concentration was estimated using the Qubit Fluorometer (Thermo Fisher Scientific). Emulsion PCR of the finally diluted library (8 pM) was carried out using the Ion OneTouch 2 system and sequencing was performed on Ion GeneStudio™ S5 System using 520 chip and 400 bp sequencing chemistry.

### 2.5. De Novo Genome Assembly and Annotation

The raw sequence reads were assessed for quality using FastQC [16]. Low-quality reads were filtered using PRINSEQ 0.20.4 where reads with an average quality score <20 and length <50 bases were discarded. Various assemblers were used for genome assembly, including the ABySS assembler [17], Velvet de novo assembler [18], SOAPdenovo2 [19], SPAdes [20], and Newbler 2.7 [21]. For the gene prediction, we used Prodigal [22], Glimmer3 [23], the NCBI prokaryotic genome annotation pipeline which uses GenemarKs + 2 s [24], and MetaGeneAnnotator [25], and finally, the genes were predicated by the NCBI prokaryotic genome annotation pipeline for further annotation. Annotation was accomplished using the Rapid Automated Annotation using the Subsystem Technology (RAST) server v.2.0 [26]. Similarly, the NCBI prokaryotic genome annotation pipeline was also used for detailed gene analysis, where the best-placed reference protein set method was used for annotation; DFAST [27] and comparative genome analysis was carried out using the PATRIC annotation tool v.3.6.9. To obtain genomic insights on pathogenicity and virulence, the virulence factors of pathogenic bacteria (VFDB) web tool was used [28]. Proteome comparison was carried out by using PATRIC’s Circos tool which uses CIRCUS plots for the analysis of genomic structural variants with sets of closely related organisms’ genomes. To determine identity between genomes, an average nucleotide identity (ANI) was calculated using the ANI/AAI matrix from the Kostas laboratory between the de novo assembled genome with closely related species. The same platform was used for the phylogenetic analysis using the neighbor-joining method from whole genome sequences [29], and the iTOl interactive tree of life was used to visualize the phylogenetic tree [30].

### 2.6. Identification of Antimicrobial Drug Resistance Genes and Biochemical Pathway

The PATRIC specialty gene tool was used to curate antimicrobial genes from the PATRIC AMR database. The Comprehensive Antibiotic Resistance Database (CARD) was used to study antimicrobial drug resistance genes present in the isolate. The Promoter 2.0 server [31] was used for finding transcription starting sites from the sequences. Predicted proteins from the genome were uploaded to Blast KOALA, which is an online automated annotation program that generates KEGG orthology identifier numbers (KO IDs). The KO ID identifications were further used to map pathways from the Kyoto Encyclopedia of Genes and Genomes (KEGG) database, where KEGG mapper tool uses the KEGG2 database.

## 3. Results and Discussion

### 3.1. Morphological Characteristics and Genome Assembly Statistics

Morphologically, the organism was found to be obligate anaerobes and forms ceramist white colonies on modified Mah et al. medium [11]. Microscopic examination showed the organism to be spore-forming Gram positive, with moderate rods, motile, and found to be singly or in pair. A total of 3,116,578 clean reads were generated after trimming and used for de novo assembly. The Newbler assembler generated a minimum of 78 contigs from the genome, with higher N50, which were further selected for whole genome analysis. The assembly results with different assembler’s packages are summarized in Table 1.

### 3.2. Comparative Genome Analysis of P. bifermentans GBRC with Closely Related Species

Species identification from the public database for molecular typing and microbial genome identification PubMLST web tool [32] predicted the taxa of the isolate was *P. bifermentans* with 100% support. Figure 1 represents genome visualization of the strain GBRC. ANI analyses are commonly used to compare bacterial strains based on whole genome sequencing [33]. Usually, ANI values between genomes of the same species are above 95%. Average nucleotide identity was found to be 98.5% with *P. bifermentans* S01 (Figure 2). The digital DNA (DNA hybridization (dDDH) type (strain) genome server), provides a genome-based delineation of species [34]. The dDDH values <70% considered as an indication of separate species. The *P. bifermentans* GBRC shows the highest dDDH percentage (82.7%) with the *P. dentum* SKVG24 T. The dDDH values with nearer species are described in Appendix A. The evolutionary distance with close species inferred with the neighbor-joining method using [35] whole genome sequences from selected genomes is shown in Figure 3; the strain GBRC shows more relatedness with the strain *P. bifermentans* S01. Figure 4 shows the percent identity across all the proteins in the compared genomes, where strain GBRC shows the highest similarity with the strain ATCC 638 and strain S01.

### 3.3. Gene Predictions and Genome Annotation

The NCBI prokaryotic genome annotation pipeline (PGAP) generated 3511 genes using GeneMarkS + 2, which were then chosen for genome annotation. After predicting the genes, genome annotation was carried out using multiple platforms (Appendix A). The salient features of the genome are depicted in Table 2. The isolate included a total of 37 genes associated with virulence factors, including four toxin-related genes. (Table 3). Other genes were mainly pertaining to adherence, regulation, and immune evasion classes. Toxin-encoding genes included *hemolysin*, *perfringolysin O* (PFO), *alpha toxin*, and *kappa toxin*. The translated protein sequence of the *hemolysin* encoding gene sequence showed 99.07% identity with *hemolysin* found in *P. bifermentans*. *kappa* toxin displayed 100% identity with *collagenase* found in *Clostridium thiosulfatireducens*. *Perfringolysin O* had a 99.60% matched identity to the thiol-activated *cytolysin* family *P. bifermentans*. Other putative virulence factors (non-toxins) found in strain GBRC include those that may play a role in adherence and colonization such as capsule, *fibronectin-binding proteins*, *flagella*, and the heat shock proteins, *Catalase-peroxidase* for stress adaptation. GroEL, for example, is a heat shock protein family chaperon discovered alongside the heat-inducible transcription repressor HrcA, as is heat shock protein GrpE [36]. Fibronectin-binding protein, capsule encoding genes, flagella, and LPS O*’*Antigen genes were among the additional adherence factors.

### 3.4. Substrate Utilization and Metabolic Pathway Analysis

As shown in Table 4, different general characteristics for strain GBRC and their closely related species were compared. Strain GBRC differs from the other three isolates in that it lacks lysine decarboxylase. Starch, gelatine, and casein hydrolysis, the ability to produce H2S and indole all seem to be characteristics of strain GBRC. The results of biochemical characterization using BioLOg are shown in (Table 5). The isolate can hydrolyze gelatine, starch, and casein and also utilize various carbon and nitrogen sources as a substrate. Genomic profiles to metabolize these substrates were also studied to confirm its substrate utilization capability (Table 5). KEGG computerizes data knowledge on pathway databases and chemical reactions that are responsible for the various cellular processes [37]. Major biochemical pathway groups include cellular processes (124), metabolism (1560), human diseases (34), genetic information processing (149), organizational system (41), environment information processing (160), and drug resistance (26), (Figure 5). *P. bifermentans* is found to damage the intestinal mucosa layer and induce inflammatory response by the invasion of the tissue in the mouse [14]. Strain GBRC is found in the uterus of the buffalo; a few specific genes are present that may damage the uterus epithelial layer and cause uterine infection. For example, the internalin A gene (inlA) mainly involved in the bacterial invasion of epithelial cells further binds to the E-cadherin and MET hepatocyte growth factor receptor and invades epithelial cells [38]. In addition, pathways related to flagellar assembly, including bacterial chemotaxis (Appendix A), two-component system, and cationic antimicrobial peptide (CAMP) resistance were found in strain GBRC. Chemotaxis has been found to have a role in biofilm development in numerous microorganisms [39,40,41] and it may guide a bacterium to swim toward nutrients (hydrophobic pollutants) adsorbed to a surface, followed by flagella attachment. Flagella have been indicated as a significant organelle responsible for bacterial adhesion, biofilm formation, and invasion into host cells [42]. In various pathogens, flagellin and/or the distally located flagellar cap protein have been reported to function as adhesins [43]. The biofilm form by pathogenic bacteria helps to sequester antimicrobial agents and makes it difficult to clear the infections. *P. melaninogenica*, *A. pyogenes*, and *E. coli* have all been reported to infect uterine linings in cattle, and have also been found to produce biofilm in uterine linings, suggesting that biofilm formation could be a major cause of persistent uterine infections [44,45].

### 3.5. Antibiotic Sensitivity

*P. bifermentans* GBRC is resistant to erythromycin and co-trimoxazole (Sulpha/Trimethoprim). In several bacteria, efflux has been discovered to be the primary mechanism of erythromycin resistance [46,47]. The fact that co-trimoxazole resistance is linked to decreased susceptibility to antibiotics from many structural families reveals the overexpression of efflux pumps capable of extruding various substrates and could be the cause of the observed resistance [48]. Tetracycline resistance genes are present in the genome, although antibiotic susceptibility testing has revealed the isolate to be susceptible to this antibiotic. Further analysis showed there to be the presence of two copies of the TetM/TetW/TetO/TetS tetracycline family resistance ribosomal protection protein in different locations of the genome. When these genes were compared with gene of closely related genera, the one Tet gene was found to be truncated of having 801 base pairs (Figure 6A), another Tet gene with 1983 bp was not found to be located under the promoter (Figure 6B). This could be the possible reason for the isolate to be sensitive to tetracycline although it has the tetracycline resistance genes. Furthermore, there are several reports on the presence of tetracycline resistance genes in the environment pathogenic bacteria as a result of horizontally-acquired gene mechanisms [49,50] which may be the reason for the presence of the TetM/TetW/TetO/TetS tetracycline family resistance ribosomal protection protein in non-functional form in various locations of the genome. The KEGG pathway identified eight genes for the vancomycin resistance from the genome (Appendix A); however, strain GBRC was found to be sensitive towards the vancomycin antibiotic. To address this question further, its pathway was studied in detail with the vanRB gene acting as a response regulator for the vanHAX gene cassette, which led to the resistance of the vancomycin. However, in the *P. bifermentans* GBRC, no significant match with the vanRB gene was found. However, the vanRB gene BLAST result shares 100% similarity with the response regulator from *P. bifermentans*, which may have led to the sensitivity of the vancomycin sensitivity to the strain GBRC.

Antibiotic resistant pathogens are responsible for the persistent endometritis in bovines due to the presence of constant bovine uterine inflammation co-infection with other pathogenic species; *Escherichia coli* or *Fusobacterium necrofurom* may increase the severity of the diseases and persistence of pathogens in the uterus [51,52]. There is evidence that *P. bifermentans* PB produces a neurotoxic that has insecticidal properties, which selectively targets anopheline mosquitoes [53]. In addition, one report revealed that P. bifermentans PAGU1678 causes intestinal damage in mouse models of UC and induces an inflammatory response by invading the tissues [13]. This is the first report on *P. bifermentans* GBRC isolated from the inflamed uterus lining of *B. bubalis*, and it has genomic features that allow it to infect the uterus endometrium lining. Inflammation of the lining can be caused by vaginal bacteria entering the uterus after the birth and causing infection within six weeks of the birth. Postpartum endometritis occurs after about 1% to 3% of vaginal births in humans, and in up to 27% of cesarean births. Retention of membranes and multiple vaginal examinations appear to increase the risk [54]. Endometritis and metritis are nowadays major causes of the decline in the fertility of bovines that causes great economic loss in dairy herds [55]. Findings from this study suggest a potential risk of infection by *P. bifermentans* GBRC and cause endometritis and metritis; this may raise an alarm for bovine health.

## 4. Conclusions

The newly isolated obligate anaerobe *P. bifermentans* GBRC from the buffalo uterus environment from the family Peptostreptococcaceae suggests a potential risk of infections in bovines. The anamnesis of *P. bifermentans* GBRC presence in the slaughtered buffalo uterine organ is unknown and might be of environment origin. We used a whole genomic sequencing platform for genetic evidence on ecological significance and niche adaptive strategies for the microorganisms befitting into the current knowledge and understanding of periodontal microbiology, the reported species appears to be a late colonizer of uterine environment and has significant virulence potential to contribute towards the causation and persistence of uterine non-specific infections.

## 5. Nucleotide Sequence Accession Numbers

The assembled whole genome is deposited at ta NBCI GenBank under the accession no. JAIRBK000000000.

## Figures and Tables

**Figure 1 animals-12-01765-f001:**
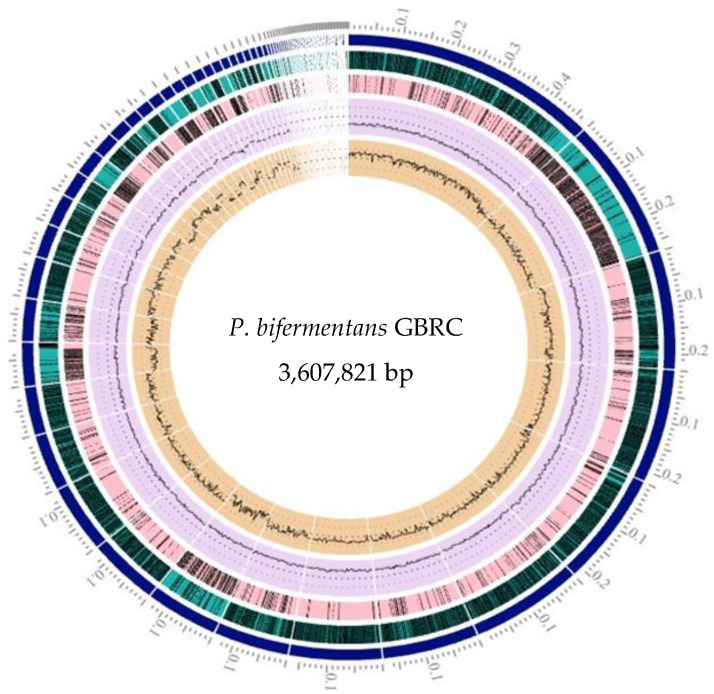
Graphical circular representation of the genome of the strain GBRC circles from interior to exterior represent GC skew, GC content, CDS on reverse strand, and CDS on forward strand.

**Figure 2 animals-12-01765-f002:**
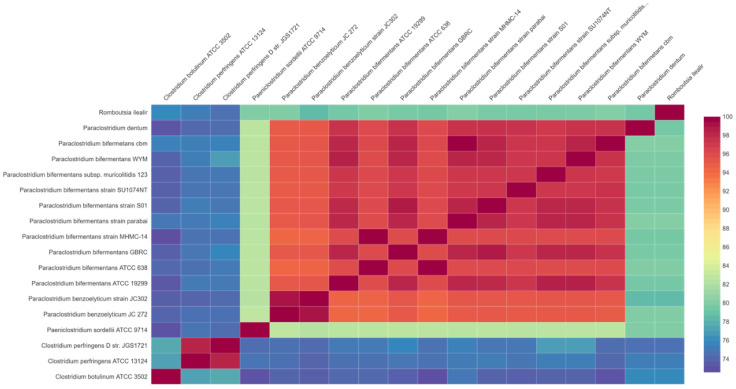
Relationship of *P. bifermentans* GBRC among other 17 closely related organisms. The average nucleotide identity (ANI) matrix was generated using ANI calculator and the enveomics collection toolbox genome was used for distance matrix calculation.

**Figure 3 animals-12-01765-f003:**
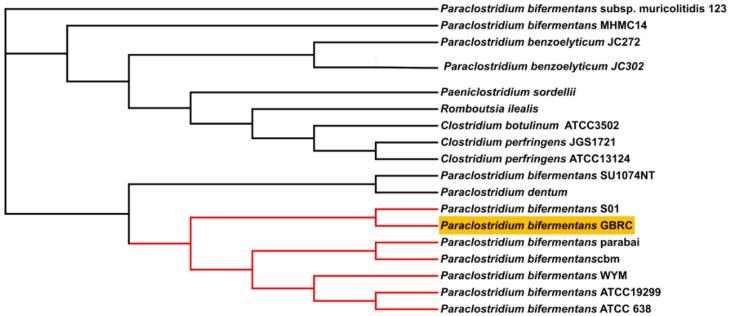
Phylogenetic tree reconstructed with the neighbor-joining method based on the whole genome sequence of the *Paraclostridium bifermentans* strains and related genus of *Paraclostridium. P. bifermentans* GBRC showing the phylogenetic relationship with the related species.

**Figure 4 animals-12-01765-f004:**
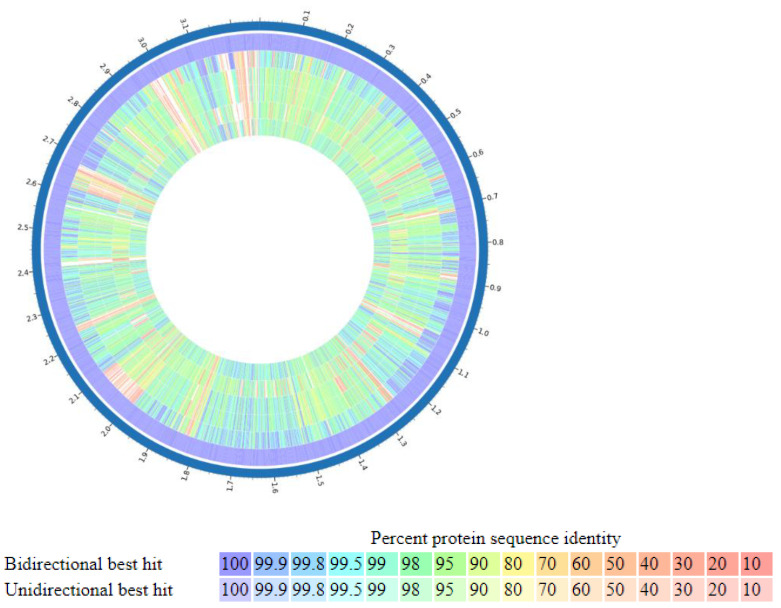
Proteome comparison: The entire proteome for each different strain is aligned against the *P. bifermentans* GBRC proteome. Track order from outside to inside is shown in the top right part of the figure where the reference strain, GBRC, is the most outside circle. The color key indicates the percentage of protein sequence similarity over the entire genome. The list of tracks, from outside to inside includes *P. bifermentans* GBRC, *P. bifermentans* strain Cbm, *P. bifermentans* strain MHMC-14, *P. bifermentans* strain DSM 14991, *P. benzoelyticum* strain JC302, and *P. bifermentans* subsp. *muricolitidis* 123.

**Figure 5 animals-12-01765-f005:**
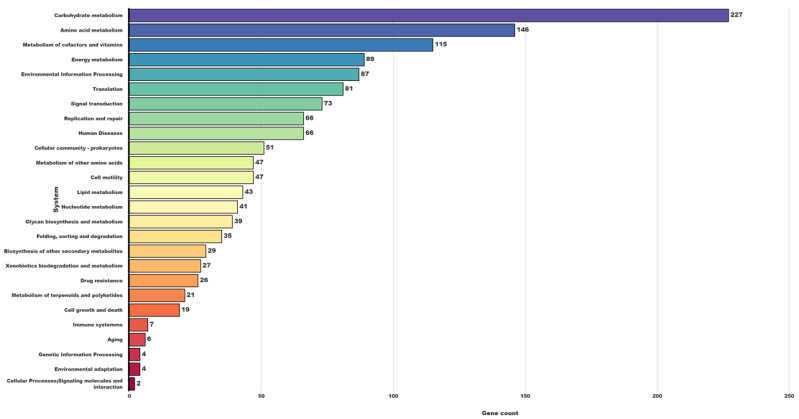
Kyoto Encyclopedia of Genes and Genomes (KEGG) classification of metabolic pathways *P. bifermentans* GBRC.

**Figure 6 animals-12-01765-f006:**
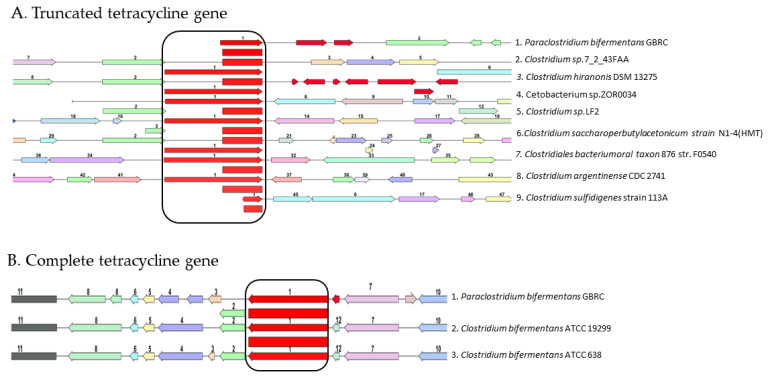
(**A**) Comparison of the truncated gene (1) TetM/TetW/TetO/TetS family tetracycline resistance ribosomal protection protein with different genera; (**B**) comparison of full gene (red colored) (1) TetM/TetW/TetO/TetS tetracycline family resistance ribosomal protection protein with different bacteria.

**Table 1 animals-12-01765-t001:** De novo genome assembly statistics for *P. bifermentans* GBRC with different assembly software.

Assembler Platform	Total Number of Scaffolds	N50 Size (bp)	Longest Contig’s Length
ABySS	73,275	240	1237
Velvet	4451	502	3135
SOAPdenovo2	3968	664	4829
SPAdes	178	165,665	517,124
Newbler 2.7	78	177,236	477,527

**Table 2 animals-12-01765-t002:** General feature of *P. bifermentans* GBRC draft genome.

Feature	*Paraclostridium bifermentans* GBRC
Domain	Bacteria
Taxonomy	*Firmicutes*; *Clostridia*; *Eubacteriales*; *Peptostreptococcaceae*; *Paraclostridium*;
	*Paraclostridium bifermentans*
Genome size	3,607,821 bp
N50 size(bp)	177,236 bp
Total Num. of contigs	78
G + C content	28.10%
Contamination	0%
Completeness	0.99
Genes	3511
CDSs (Total)	3441
Genes (Coding)	3.395
Genes(RNA)	70
rRNAs	3, 3, 5 (5 S, 16 S, 23 S)
Complete rRNAs	3 (5 S)
Partial rRNAs	3, 5 (16 S, 23 S)
tRNAs	55
ncRNAs	4

**Table 3 animals-12-01765-t003:** Virulence factors present in the *P. bifermentans* GBRC genome.

Virulence Class	Virulence Factors	Related Genes	Predicated Gene ID
Adherence	Fibronectin-binding protein	fbpA/fbp68	gene_12_113
	GroEL	groEL	gene_11_64
	Flagella (*Pseudomonas*)	flip	gene_3_40
	LPS O-antigen (*P. aeruginosa*)		gene_1_129, gene_1_130
	Listeria adhesion protein	Lap	gene_7_23
Regulation	CheA/CheY (*Listeria*)	cheY	gene_3_167
	LisR/LisK (*Listeria*)	lisR	gene_19_8, gene_8_135
	Sigma A (*Mycobacterium*)	sigA/rpoV	gene_9_93
Toxin	Alpha-toxin	Plc	gene_18_29
	Hemolysin		gene_11_116
			gene_8_36
	Kappa-toxin (collagenase)	colA	gene_11_57
	Perfrigolysin O (theta-toxin/PFO) /botulinolysin	pfoA	gene_6_144
	Cytolysin	cylR2	gene_10_57
Antiphagocytosis	Capsular polysaccharide	rmlB	gene_12_15
	Capsular polysaccharide	wcaJ	gene_8_159
	Capsule		gene_3_150
Cell surface Components	Trehalose-recycling ABC transporter	sugC	gene_13_45
Immune evasion	Capsule	cps4I	gene_26_21
			gene_8_154
	LPS	acpXL	gene_26_5
	Polysaccharide capsule		gene_8_145
			gene_8_146
			gene_8_155
			gene_8_161
	Polysaccharide capsule	epsE	gene_12_3
		manA	gene_12_22
Iron uptake	Heme biosynthesis	hemB	gene_25_23
	Periplasmic binding protein	vctC	gene_13_103
Nutritional virulence	Pyrimidine biosynthesis		gene_11_52
Secretoin system	T6SS-II		gene_29_17
Serum resistance and immune evastion	Capsule		gene_12_130
	LPS	wbtE	gene_8_165
	LPS	wbtF	gene_8_166
	LPS	wbtI	gene_3_25

**Table 4 animals-12-01765-t004:** General characteristics of Taxa: 1, *P. bifermentans* JCM 1386T [12]; 2, P. benzoelyticum JC272T [13]; 3, *P. bifermentans* subsp. Muricolitidis [12]; 4, *P. bifermentans* GBRC +, positive; −, negative.

Characteristic	1	2	3	4
Gram-stain	Gram-Positive	Gram-Positive	Gram-Positive	Gram-Positive
Shape	Rod Shaped	Rod Shaped	Rod Shaped	Rod Shaped
Sporulation	Central to Terminal	Terminal	Terminal to Sub terminal	Terminal
Motility	Motile	Motile	Motile	Motile
Obligate Anaerobes	Obligate Anaerobe	Obligate Anaerobe	Obligate Anaerobe	Obligate Anaerobe
Catalase	−	−	−	−
Oxidase	−	−	−	−
Indole	+	−	+	+
H2S production	+/−	+	−	+
Starch hydrolysis	−	+	−	+
Gelatin hydrolysis	+	+	+	+
Nitrate reduction	−	−	−	−
Lysine Decarboxylase	+	+	+	−
Casein	+	+	+	+

**Table 5 animals-12-01765-t005:** Substrate utilization profile of the *P. bifermentans* GBRC as obtained by AN-Biolog^®^ Plate as well as genomic approach.

Classification		Substrate	Predicted KEGG Pathway ID
System	Subsystem		
Carbohydrates	Ketose	d-fructose	map00051, map00052, & map00500
	Monosaccharide	α–d-Glucose	map00010, map00052, & map00520
		d-Mannose	map00051
		α–d-Glucose 6 phosphate	map00010, map00030, & map00052
		Galactose	map00052
	Hexosamine	N-Acetyl-d-Galactosamine	map00052
	Sugar Alcohol	Sorbitol	map00051
	Nucleoside	Inosine	map00230
Amino acids		l-Asparagine	map00250, & map00460
		l-Glutamine	map00250
		l-Threonine	map00260
		l-Methionine	map00270
		l-Serine	map00260, & map00270
		l-Proline	map00330, & map00470
		l-Phenylalanine	map00360

## Data Availability

The data will be available with the corresponding author upon request.

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
