# Peer review of "In-Depth Analysis of an Obligate Anaerobe Paraclostridium bifermentans Isolated from Uterus of Bubalus bubalis"

_animals, 2022, doi:10.3390/ani12141765_

Round 1

Reviewer 1 Report

The paper entitled "In-depth analysis of an obligate anaerobe Paraclostridium bifermentans isolated from uterus of Bubalus bubalis" represents a well structured manuscript on the characterization of a novel species isolated from uterine environment in a regularly slaughtered buffalo.
The study design for characterization is clearly described and well detailed. 
However, some concern rises about the overall scientific soundness: the study is based on a single specimen found at the slaughterhouse and no other information about the live animal is provided. I can undestrand that retrieving data back is difficult and that probabily information such as: parity, days in milk, concurrent diseases... are no longer available. But clinical inspection of the organs could be useful to readers: were follicles/corpora lutea/cysts present on the ovaries? what about the dimensions: was that a postpartum animal? was the uterus completely involuted or still in the involution process? were there any iatrogenic lesions, maybe caused by wrong insemination method? could the bacteria entered the uterus through contaminated insemination instruments? These information could be useful to understand whether the novel characterized bacteria could be pathogenic or opportunistic, and drive future investigations.

I assigned the paper to "major revision" in order to let the Editor decide whether to suggest a re-classification of the paper: as a personal opinion, since the study is based on a single specimen found at the slaughterhouse, maybe this paper could be considered closer to a case-report?

Minor comments:
Line 74: buffaloes
Line 86: were followed
Line 267-268: I think one word is missing "Inflammation of the uterine lining"?
Line 270: The clinical term "prolonged rupture of membranes" needs clarification: do you mean "premature rupture of membranes"? Or "retention of foetal membranes"?
Line 273-275: according to the previous consideration, this report is based on only one specimen collected after slaughter. In my own opinion, conclusions should be drown more cautiously. Although genes for expression of pathogenicity factors were found, no anamnesis or clinical inspection of the organs are available and the presence of the bacterial population could also be due do iatrogenic damage (wrong insemination method/contaminated insemination instruments). These hypothesis should be discussed into the section.

Author Response

Authors comments to the reviewer:

We would like to thank the reviewer for their helpful comments, critics and questions. Please see our specific comments and descriptions of changes below. As a result of these changes, we feel that the manuscript has improved considerably - thank you!

All changes made to the manuscript have been highlighted in yellow.

Best regards,

Vishal

Reviewer 2 Report

The article is an original study conducted at a high methodological level. The purpose of the work is well understood. The sampling procedure, primary isolation, description and testing of the bacterial culture are described quite clearly and indicate the representativeness of the analyzed material. The technique of molecular genetic studies of the obtained P. bifermentans isolate is described in sufficient detail and gives an idea of ​​the high level of research. The results of the work are presented in a clear language. This concerns the genomic characteristics of the isolate, its comparison with closely related bacterial strains, as well as the use of the substrate and the metabolic pathway. Tables and figures are quite informative and well illustrate the results of the study. In general, the study is of great scientific and practical importance.

Comments.

1) Key words do not adequately reflect the objects of research. For example, they don't mention bacteria, genomic research.

2) The introduction shows the significance of the problem. However, the mention of bacteria that can cause diseases of the buffalo uterus is superficial. The authors write about opportunistic pathogens. Is it possible to say which bacteria are most often found in the environment of the buffalo uterus, are there specialized pathogens among them?

3) The discussion of the results (Section 3.4.) is somewhat sparse. For example, the authors mention biofilms in which bacteria exist. However, nothing is said about the possible involvement of P. bifermentans in biofilms formed in the uterus of animals with endometritis and metritis.

4) It would be important to know the opinion of the authors about the possible interactions of the identified bacterium with other microorganisms that develop in the uterus of sick animals. Obviously, this is a mixed infection. The question arises about the place of P. bifermentans in the bacterial community in the uterus of buffaloes?

5) It would probably be useful to say about the possible ways of settling this bacterium.

6) Since this is the first report of the discovery of P. bifermentans as a potential risk of infection in cattle, it would be interesting to know the opinion of the authors on the prospects for further steps in this study.

7) The authors recommend paying attention to the risk of occurrence of this bacterium. The question arises is it possible to recommend preventive or other methods to limit the risks of animals being affected by this pathogen?

Author Response

(The authors gave the same response as above.)
